# Deterrence by Collective Punishment May Work against Criminals but Never against Freedom Fighters

## Friedel Bolle

Department of Business and Economics, European University Viadrina, 15230 Frankfurt (Oder), Germany; bolle@europa-uni.de

**Abstract:** The main goal of collective punishment (CP) is the deterrence of future "wrong-doing" by freedom fighters or terrorists, protesters against an authoritative government, polluters, students playing pranks on their teacher, football teams lacking enthusiasm, or soldiers showing cowardice to the enemy. CP could consist of the lockout of workers, additional training units for football teams, increased control of athletes and firms, up to the shooting of fellow villagers of assassins. I investigate two classes of problems. In one class, resistance against an authority is individually costly, but enough resistance can be successful (the production of a public good, for example, higher wages after a strike). In the other case, "resistance" is individually profitable (a criminal activity as pollution) and enough "resistance" produces a public bad. We find that, in the first situation, the announcement of CP never decreases the level of resistance. In the second situation, CP can be successful.

**Keywords:** collective punishment; public good; public bad; resistance





## 1. Introduction

There are countless situations where collective punishment (CP) is or could be applied. It is applied in situations where culprits have been caught as well as in situations where they are unknown. The main goal of CP is deterrence of future "wrong-doing" by freedom fighters or terrorists, polluters, students playing pranks on their teacher, football teams lacking enthusiasm, soldiers showing cowardice to the enemy, etc. CP is often applied without calling it CP. The answer to a strike by some may be the lockout of all workers. The counterfeiting of products or labels may be followed by a boycott of an industry or a country. The misuse of government subsidies by some firms may cause the cancelling of the whole program. Even the "Zero-Tolerance Policy" may be regarded as a form of CP [1].

In this investigation, a principal wants a group of agents to follow certain rules of behavior (a regime). The members of the group are motivated to resist against the regime, either because of personal advantages (criminal behavior) or for the best of their group (social behavior). After observing a certain level of misconduct, the principal applies CP, i.e., all the members of the group suffer from his punishment. The research question is: *Which height of the punishment minimizes the probability of misconduct?* CP is modelled as an institution for the management of social dilemmas. It aims to suppress or foster cooperation, thus increasing or decreasing the conflict between individual and social interests.

### 1.1. The Fallacy of the Higher Effectiveness of Higher Punishment

There are many claims that a severe CP reliably deters criminal wrongdoers, for example: *If you know that one member of a group has stolen $10, fine everyone $20* [2]. This may work, ex ante or ex post, with communication in the group. However, with independently acting group members, it is not necessarily convincing. Subsequently, a player's best reply to the automatic execution of CP after one act of stealing depends on the probability with which she expects her co-players to steal. The members of the group play a Stag Hunt

game with two pure strategy equilibria: All remain honest or all steal. Our environment is richer than this simple example, but, again, we have to deal with multiple equilibria and the necessity to apply equilibrium selection. We find that increasing punishment does not necessarily imply decreasing stealing.

### 1.2. Substitute or Amplification of Individual Punishment

In some cases, CP is applied when the culprit is known but not caught; but, even if the culprit is caught, CP may be applied because it extends the pain of punishment over the limit, which is possible by individual punishment. Imagine that, instead of you, your child has to suffer corporal punishment, hunger, or the loss of its home. It is similar when other family members or fellow villagers are affected. A culprit may feel even worse if only the other (innocent) members of a group are punished, but not him. This motive for CP existed in the Roman hostage taking from the nobility of defeated people. It is typical for many cases of revenge taking as vendettas between families or clans, it is the rationale for bombs against civilian targets (by terrorists or regular armies), and it is a motive for Israel's destruction of houses that are owned by the families of Palestinian assassins.

### 1.3. Moral and Legal Issues

The literature on CP is mainly concerned with moral and legal aspects. Authors discuss questions of collective guilt, collective responsibility, human rights, international law, national criminal law, and philosophical and religious principles. The discussion often concentrates on certain examples from the Bible and ancient Greek dramas (often concerned with God's CP), or examples of modern resistance fighters or terrorists (depending on the viewpoint). Israel's destruction of the houses of assassins' families or fellow village people is central for many contributions (e.g., [3]), although there are more examples with similar politics [4]. Related to a moral discussion is the investigation of social efficiency of CP [5]. However, this investigation is not concerned with moral or legal issues. The central concern is the modeling of situations where CP could be applied and the question of which effect a credible threat of CP has. Moral arguments are only raised in the discussion of some details of the model as, for example, a maximum height of CP.

### 1.4. Selected Literature

Heckathorn [6] investigates a model situation with two groups, inter-group punishment and strong intra-group relations of the players. Kritikos [7] suggests CP for environmental problems (the negative frame of public goods provision) where polluters cannot be identified individually but collectively (for example, because they are located in the basin of a polluted river). *Experiments:* Behavioral economics teach us that behavior derived from theoretical models often does not match behavior in the field or in the laboratory. For example, we may conclude from Wang et al.'s [8] Prisoner's Dilemma experiments that adding an (irrelevant) alternative to resisting or complying with the authority may considerably change behavior. Decker et al. [9] find increasing contributions to a public good after the introduction of CP for insufficient aggregate contributions. Dickson [10] finds that trying to prevent the production of a public bad by CP is counterproductive. Neither does he confirm Decker et al.'s [9] result for the positive frame, at least not in the long-run. Gao et al. [11] find that collective punishment is more effective in promoting cooperation (producing a public good) than collective rewards. As we have emphasized above, CP will be more effective if the wrongdoer has an empathic connection with the punished group. [Complementary to this expectation, Pereira and van Prooijen [12] show that people are less reluctant to carry out CP against an "entitative" group (for example, a biker gang), because members of such groups are perceived to be more similar and interchangeable. Falomir-Pichastor et al. [13] and Pereira et al. [14] connect the readiness to carry out CP between groups with the decision structure in these groups. The effectiveness of extant CPs has rarely been investigated. We will come back to this question in the conclusion. *Related research.* There is a loose connection with discussions of the effectiveness of

individual punishment (theory starting with the seminal work of Becker [15]; experiments with an environment of producing a public good starting with and Fehr and Gächter [16], in particular when there are strong external effects of asocial behavior. Even without CP, such investigations have a certain relation to this investigation if individual punishment relies on or induces network effects. For example, punishment may need the backing by the group (in principle [17–19], or as at least some consensus [20], or has consequences for network reciprocity [21].

*1.5. CP against the Costly Production of a Public Good*

Let us imagine protests against an authoritative government. The protesters bear individual costs in the form of time and, in cases of attempts by the authority to suppress protests, also in terms of potential injuries, confinement, and so on. However, if there are enough protesters, the government may comply, partly or completely, with the requirements of the protesters (e.g., the yellow vest movement in France). On the other hand, the protests may trigger CP, possibly concerning the families of protesters and often the general population, as, for example, by a curfew, restrictions to the access of social media and frequent police controls with arbitrary accusations and arrests (protests against the governments in Belarus, Russia, Myanmar, and Hong Kong; strikes; actions against occupiers). The crucial determinants for this type of situations are individual costs of the protesters (strikers, assassins) and the production of a public good for the group that they represent, provided enough join the protests. For a clear differentiation of cases, both attributes are important. My investigation of this case has a surprising result: *the authority cannot decrease the level of resistance by announcing CP.* Under the expectation that others resist with sufficiently high probability, agents can avoid CP only by "success"; therefore, a higher CP motivates higher resistance in spite of its costs. However, this plausible argument needs confirmation by a formal investigation.

*1.6. CP against the Profitable Production of a Public Bad*

Imagine a number of firms that can save costs by violating environmental laws, catching quotas for fish, or safety standards. A public bad is produced if too many of them pollute, catch too much fish, or neglect safety standards. Waterworks have to add additional filters or costly purification procedures or take their water from distant sources, the fish population may break down, and contributions to accident and health insurance may increase for all firms. CP may take the form of increased corporate taxes, mandatory expensive automatic controls, and a public shaming of a whole industry, which may decrease the good-will capital of all firms and severely damage brands. The crucial structure of this case, namely individual gains and the production of a public bad, can also be found in cases of cowardice of soldiers, punished in the Roman army often by the procedure of decimation, i.e., every tenth man was executed by his own comrades. In this case, the result of my theoretical investigation is less clear: At least if (saved) costs are small when compared to the (negative) externalities of a public bad, the application of *CP improves compliance* with the law or standards of behavior. Contrary to the case of CPGs, an increased CP cannot be compensated by a higher probability of success. If the advantages of "resisting" are large compared with the negative externality of "success", then, often, *complete resistance is the equilibrium,* which applies, even under high CP.

In the following Section, I will formalize the CP game and derive its mostly multiple equilibria. In Section 3, the equilibrium selection is applied with a non-formal summery in Section 3.3. In Section 4, the results are discussed.

## 2. The CP Game and the Equilibria of CP(x) Subgames

The examples of CP are rather different and often emotionally loaded. Therefore, I introduce a neutral principle-agent description for the general game.

**Definition 1.** (*The CP game*): *There is a principal* (*player* 0) *and n agents* (*player set* $N = \{1, \ldots, n\}$).

- In Stage 1, the principal determines his rules of behavior (regime), the trigger, and the height $x \in [0, M]$ of CP. We do not specify the regime and we assume that the trigger of CP is one resisting agent. Hence, only $x$ has to be determined.
- In Stage 2, all of the agents simultaneously decide whether to comply with the regime or resist against it. The subgames of stage 2 are called CP(x).

Let us assume that $j = j(N)$ agents resist. These agents bear individual costs c. The result of the players' decision is:

- If $j = 0$, the utility of all agents is $U_i = 0$.
- If $0 < j < k$, then the group suffers from CP with utilities $U_i = -c - x$ if $i$ had resisted and $U_i = -x$ if $i$ had complied.
- If $j \geq k$, all agents enjoy the additional utility G, i.e., $U_i = G - c$ if $i$ had resisted and $U_i = G$ if $i$ had complied.

The goals of the players are:

- The principal wants to maximize the probability of compliance.
- Agent $i$ wants to maximize her expected utility

$$U_i = G * prob(j \geq k) - x * prob(0 < j < k) - c * p_i, \tag{1}$$

with $p_i = $ *probability that i resists*.

This definition shows the close relation of the CP game with Threshold Public Good games. Contrary to such games, CP has two thresholds with possibly opposite effects. CP is triggered when a certain number of players resist (here set to 1); the "success" of resistance is triggered when at least k players resist. Theoretical analyses of Binary Threshold Public Good games, including Voting Games, are Bolle [22] and Bolle and Spiller [23]; recent experimental studies of Threshold Public Good games are Bolle and Otto [24], Wang et al. [25], and Bolle and Spiller [23]. See these articles for earlier experimental work on Threshold Public Goods, in particular the Volunteer's Dilemma and the Stag Hunt game.

The definition above has constant "costs" *c* and "benefits" *G*. We assume that the principal's regime breaks down when at least *k* the of the *n* players resist. CP is no longer applied if a protest is successful or an occupier is expelled. For criminal activities, in severe cases, the rule of law is abandoned if at least *k* agents resist. In minor cases (e.g., denying people access to a park for a certain time, after some of them entered the lawn), the rules of conduct change if too many agents resist. The negative externality substitutes CP.

The assumption that CP will not be applied if at least *k* agents resist is not always plausible; the hostages of the Roman Empire could be killed, whether or not the nobility of a distant country revolted successfully. Other cases of hostage taking or posting [26,27] are not connected with the production of a threshold public good and are, therefore, neither a CP game in the sense of Definition 1.

In the following, we denote $p = (p_i)_{i=1,\ldots,n}$ as the vector of mixed strategies. $p_i = 0$ and $p_i = 1$ are identified with the respective pure strategies. $p_{-i} = (p_i)_{i=1,\ldots,i-1,i+1,\ldots,n}$. $j(T)$ denotes the number of resisting agents in the set of agents $T$.

$$prob(j(N) \geq k) = p_i * prob(j(N - \{i\}) = k - 1) + prob(j(N - \{i\}) \geq k), \tag{2}$$

$$prob(j = 0) = (1 - p_i) * prob(j(N - \{i\}) = 0), \tag{3}$$

$$prob(0 < j < k) = 1 - prob(j = 0) - prob(j(N) \geq k), \tag{4}$$

$$\partial U_i / \partial p_i = prob(j(N - \{i\}) = k - 1) * G,$$
$$-(prob(j(N - \{i\}) = 0) - prob(j(N - \{i\}) = k - 1)) * x - c, \tag{5}$$

**Lemma 1.** $\partial U_i / \partial p_i$ *depends only on* $p_{-i}$. *Therefore, if* $\frac{\partial U_i}{\partial p_i} > 0$ *(*$\frac{\partial U_i}{\partial p_i} < 0$*) then* $p_i = 1$ *(*$p_i = 0$*) is the unique best reply of player i. A strictly mixed strategy equilibrium requires* $\frac{\partial U_i}{\partial p_i} = 0$ *for all i.*

**Proof.** (5). □

For the sake of simplicity, we have assumed symmetric agents and symmetric subgames CP(x). In a symmetric game, only symmetric equilibria should apply. Subsequently, equilibrium requires $p_j = \pi$ for all $j$. We define $U_i'(\pi) = \frac{\partial U_i}{\partial p_i}(p_j = \pi$ for all $j)$.

$$prob(j(N - \{i\}) = k - 1) = \binom{n-1}{k-1} * \pi^{k-1}(1 - \pi)^{n-k}, \tag{6}$$

$$prob(j(N - \{i\}) = k - 1) = (1 - \pi)^{n-1}, \tag{7}$$

$$U_i'(\pi) = (x + G) * \binom{n-1}{k-1} * \pi^{k-1}(1 - \pi)^{n-k} - x * (1 - \pi)^{n-1} - c, \tag{8}$$

In the case $k = 1$, if $0 \leq c/G \leq 1$, then $U_i'(\pi) = 0$ has exactly one solution

$$\pi = \eta = 1 - \left(\frac{c}{G}\right)^{1/(n-1)}, \tag{9}$$

which is independent of $x$. CP is never applied, and we can exclude this case from the further discussion. Nonetheless, the probability $\eta$ plays a role in the description of other cases. In order to not get lost in too many subcases, from now on we are only concerned with not too extreme $k$. The cases left out do not offer qualitatively different results, but they require special discussions, at different points of the investigation.

**Assumption 1.** $2 < k < n - 1$.

**Lemma 2.**
1.  *If* $G < c$ *and* $c > 0$, *then* $\pi = 0$ *is a dominant strategy, independent of* $x$.
2.  *If* $c < G$ *and* $c < -x$, *then* $\pi = 1$ *is a dominant strategy.*

**Proof.** Ad 1: From (8) follows $U_i'(\pi) < G - c < 0$ for all $\pi$. Ad 2: If $x + G > 0$, (8) implies $U_i'(\pi) > x + G - x - c = G - c > 0$, otherwise $U_i'(\pi) > -x - c > 0$ for all $\pi$. □

In case 1, the principal need not threaten with CP because all agents comply already for $x = 0$. In case 2, he cannot threaten with low x, because, then, $\pi = 1$ is a dominant strategy. Case 2 with $c < G < 0$ is anyway of little relevance because the negative externalities $G$ are smaller than the negative costs (positive utilities) of "misbehaving", i.e., misbehaving is efficient. Because of this argument and Lemma 2, we will now concentrate on cases $0 < c < G$ and $G < c < 0$. These are the relevant cases and cases where we have to cope with multiple equilibria and equilibrium selection.

**Definition 2.** *A CP game with parameters* $0 < c < G$ *is called CP against a Public Good (CPG). A CP game with* $G < c < 0$ *is called CP against a Public Bad (CPB).*

Let us now visualize the condition for mixed strategy equilibria. $U_i'(\pi) = 0$ implies

$$\begin{aligned} q(\pi)\binom{n-1}{k-1} * \pi^{k-1}(1 - \pi)^{n-k} \\ = \frac{x}{x+G} * (1 - \pi)^{n-1} + \frac{c}{x+G} =: r(\pi, x), \end{aligned} \tag{10}$$

$q(\pi)$ is the probability that an agent is decisive for the "success" of resistance, i.e., the probability that exactly $k - 1$ other agents resist. $r(\pi, x)$ measures an agent's opportunity

costs of resisting in relation to her benefits $x + G$. $q(\pi)$ is a unimodal function with a maximum at $\frac{k-1}{n-1}$. $r(\pi,.)$ is a decreasing, always convex or always concave, function. All of the intersections of $q(\pi)$ and $r(\pi, x)$ represent mixed strategy equilibria of the CP(x) subgame. For increasing $x$, $r(\pi, x)$ rotates around the point $(\eta, c/G)$ with $\eta$ defined by (9). The limiting functions are $r(\pi, 0) = \frac{c}{G}$ and $r(\pi, \infty) = (1 - \pi)^{n-1}$. In the CPG game, the rotation is clockwise; in the CPB game, it is counter-clockwise. In Figure 1, the blue lines are examples for $r(\pi, x)$ in the CPB game, the limiting functions are red. The $r(\pi, x)$ of the CPG game with positive parameters $c$ and $G$ would fill the empty space between $r(\pi, 0)$ and $r(\pi, \infty)$ when the rotation is clockwise.

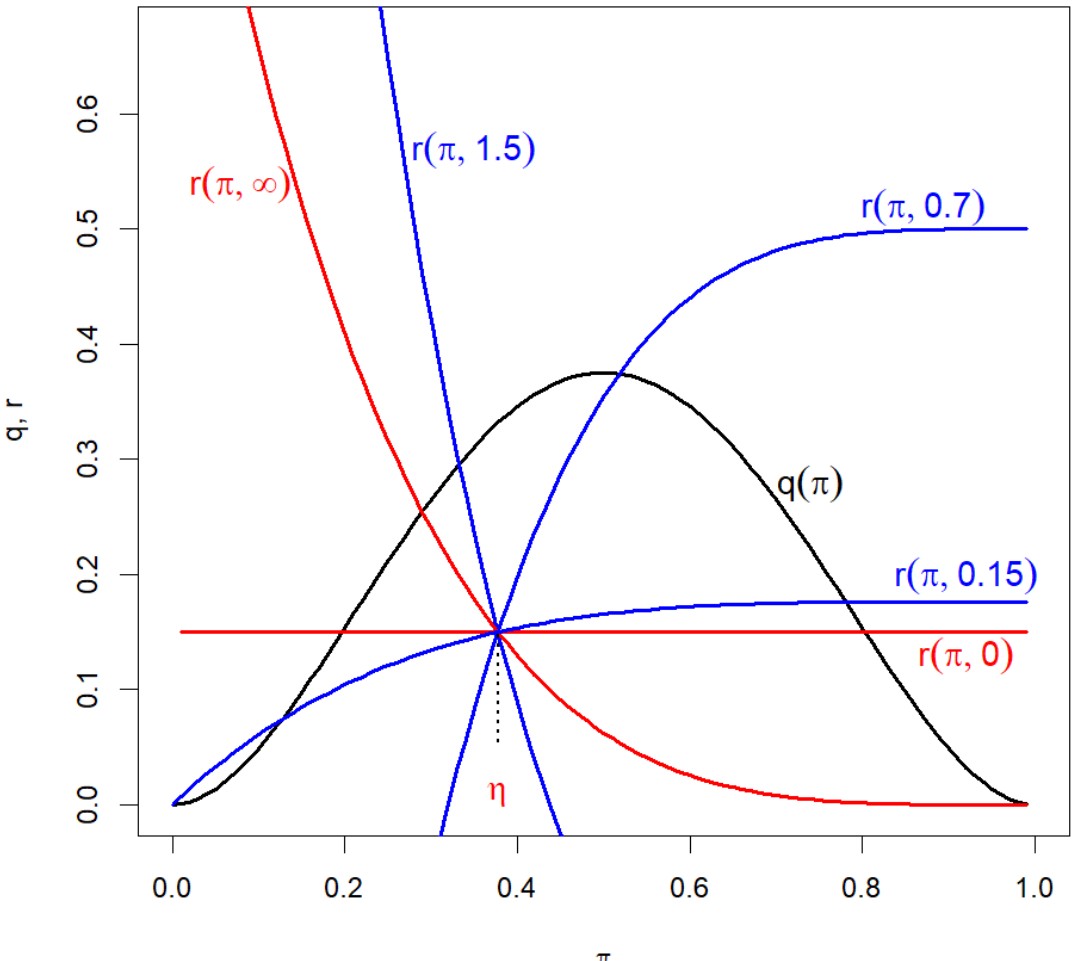

**Figure 1.** The equilibria of the CPB game as intersections of the black curve $q(\pi)$ and the colored curves $r(\pi, x)$. The extreme cases are no CP, i.e., $x = 0$, and CP with $x \to \infty$ (red curves). Blue curves are with intermediate $x$. Parameters of the figure: $n = 5$, $k = 3$, $c = -0.15$, $G = -1$.

Figure 1 illustrates a situation where the rotation point $(\eta, c/G)$ is below $q(\pi)$. Therefore, there are two mixed strategy equilibria $\pi^1(0) < \pi^2(0)$ for $x = 0$. The rotation point can take other positions, but it can never lie "to the right" of the curve $q(\pi)$.

**Lemma 3.** *If the CP(0) subgame has two mixed strategy equilibria $\pi^1(0) < \pi^2(0)$ (or, a limiting case, one mixed strategy equilibrium $\pi^1(0) = \pi^2(0)$), then $\pi^2(0) > \eta$.*

**Proof.** See Appendix A. □

**Proposition 1.**

1. *In the CPG game, $\pi = 0$ is always an equilibrium and $\pi = 1$ never. In the CPB game, $\pi = 1$ is always an equilibrium and $\pi = 0$ if and only if $x + c > 0$.*
2. *In the CPG game, $U_i'(\pi)$ is a unimodal function. Either $U_i'(\pi) = 0$ has no solution and $U_i'(\pi) < 0$ for all $\pi$ or it has two solutions $\pi^1 \le \pi^2$ and $U_i'(\pi) > 0$ for $\pi^1 < \pi < \pi^2$ and $U_i'(\pi) \le 0$ otherwise.*
3. *In the CPB game, there are three different cases.*

- *If $x < -c$, then $U_i'(\pi)$ is a unimodal function. $U_i'(\pi) = 0$ has either no solution with $U_i'(\pi) > 0$ for all $\pi$ or it has two solutions $\pi^1 \le \pi^2$ with $U_i'(\pi) < 0$ for $\pi^1 < \pi < \pi^2$ and $U_i'(\pi) \ge 0$ otherwise.*
- *For $-c < x < -G$, $U_i'(\pi)$ has either no or two local extrema. $U_i'(\pi) = 0$ has either one solution $\pi^*$ with $U_i'(\pi) < 0$ for $\pi < \pi^*$ and $U_i'(\pi) \ge 0$ otherwise; or, it has three solutions $\pi^1 < \pi^* < \pi^2$ with $U_i'(\pi) < 0$ for $\pi < \pi^1$ and $\pi^* < \pi < \pi^2$ and $U_i'(\pi) \ge 0$ otherwise.*
- *If $x > -G$, then $U_i'(\pi) = 0$ has exactly one solution $\pi^*$ with $U_i'(\pi) < 0$ for $\pi < \pi^*$ and $U_i'(\pi) > 0$ for $\pi < \pi^*$.*

**Proof.** See Appendix A. □

Even in cases with a unique symmetric mixed strategy equilibrium, there is always at least one additional symmetric pure strategy equilibrium. Therefore, we have to apply equilibrium selection, formally or informally, in order to derive policy conclusions.

## 3. Equilibrium Selection in the CP(x) Game

Among the large number of suggestions for equilibrium selection, according to my view, three are eminent and more often applied than others. First, there is the Linear Tracing Procedure of Harsanyi and Selten [28], second, the Global Games selection of Carlson and van Damme [29], and, third, the limit of Quantal Response Equilibria (QRE), proposed by McKelvey and Palfrey [30]. All three suggestions describe a continuous path of equilibria, starting with maximal noise (incomplete information) or with decisions with equal probability for every decision option. Along the path, noise vanishes or decisions respond increasingly to the structure of the original game. If the path is unique then its endpoint (necessarily a Nash equilibrium of the original game) constitutes the selected equilibrium. However, the Global Games selection does not suit the CP(x) game. Frankel et al. [31] show that, for non-concave games, the Global Games selection may depend on the distribution of noise. Therefore, because CP(x) is not a concave game, we should not apply the Global Games selection. In the following, we concentrate on equilibrium selection by the two other suggestions.

### 3.1. The Limit of Logistic Quantal Response Equilibria

QRE are derived from the assumption that players apply a mixed strategy with the probabilities of actions ordered according to the utilities that these actions gain against the choices of the other players. In the frequently used logistic version and applied to CP(x), an agent *i* chooses to resist with probability

$$p_i = \frac{\exp(\lambda * U_i(resist))}{\exp(\lambda * U_i(resist)) + \exp(\lambda * U_i(\,comply))} \\ = \frac{1}{1 + \exp(\lambda * [U_i(\,comply)) - U_i(resist)])}, \tag{11}$$

$\exp(.)$ is the exponential function and $0 \le \lambda < \infty$ is a "precision parameter". For $\lambda$ close to 0, all $p_i$ are close to ; for $\lambda \to \infty$, the probabilities of all suboptimal replies converge to 0. The path of equilibria starts with $\lambda = 0$ and the symmetric equilibrium $p_i = \pi = 1/2$.

For $\lambda > 0$, there are symmetric equilibria $p_i = \pi$ that are computed from (11). From (1)–(4) and (8) follows

$$U_i(comply) - U_i(resist) = -U_i{}'(\pi) = (G+x)[r(\pi, x) - q(\pi)],\qquad(12)$$

Because of (11) and (12), the symmetric QRE is determined by

$$f(\pi, \lambda, x) := \exp(-\lambda * U_i{}'(\pi)) = \frac{1-\pi}{\pi} g(\pi),\qquad(13)$$

The shape of the function $f(\pi, \lambda, x)$ follows from the characterization of $U_i{}'(\pi)$ in the previous section. There is a generically unique continuous path of QRE equilibria from $\lambda = 0$ to an arbitrarily large $\lambda$ [30]. The limit of this path is a Nash equilibrium of the CP(x) game.

**Proposition 2.** *(Selection according to limits of QRE):*
1.　*Let us assume adjacent mixed strategy equilibria $0 \le \pi^1 < \pi^2 \le 1$ and $\frac{1}{2} \in (\pi^1, \pi^2)$. If $U_i{}'(\pi)$ is negative in the interval $(\pi^1, \pi^2)$, then $\pi^1$ is selected, if $U_i{}'(\pi)$ is positive in this interval, then $\pi^2$ is selected.*
2.　*If, including pure strategy equilibria, the smallest equilibrium is $\pi^1 > 0$ and $\frac{1}{2} \in (0, \pi^1)$, then $\pi^1$ is selected. If the largest equilibrium is $\pi^2 < 1$ and $\frac{1}{2} \in (\pi^2, 1)$, then $\pi^2$ is selected.*

**Proof.** Ad 1. Figure 2 describes a situation with positive $U_i{}'(\pi)$ in the interval $(\pi^1, \pi^2)$ and $U_i'(\pi^1) = U_i'(\pi^2) = 0$. For increasing $\lambda$, the function $f(\pi, \lambda)$ converges to zero for all $\pi$ and the QRE equilibrium probability $\pi^{QRE}(x, \lambda)$ moves along the curve $g(\pi)$ to $\pi^2$. For negative $U_i{}'(\pi)$, all of the $f(\pi, \lambda)$ values increase infinitely and $\pi^1$ is selected. Ad 2. Leaving aside non-generic cases, $\pi = 0$ is an equilibrium if and only if $U_i'(0) < 0$, and $\pi = 1$ is an equilibrium if and only if $U_i'(1) > 0$. $\square$

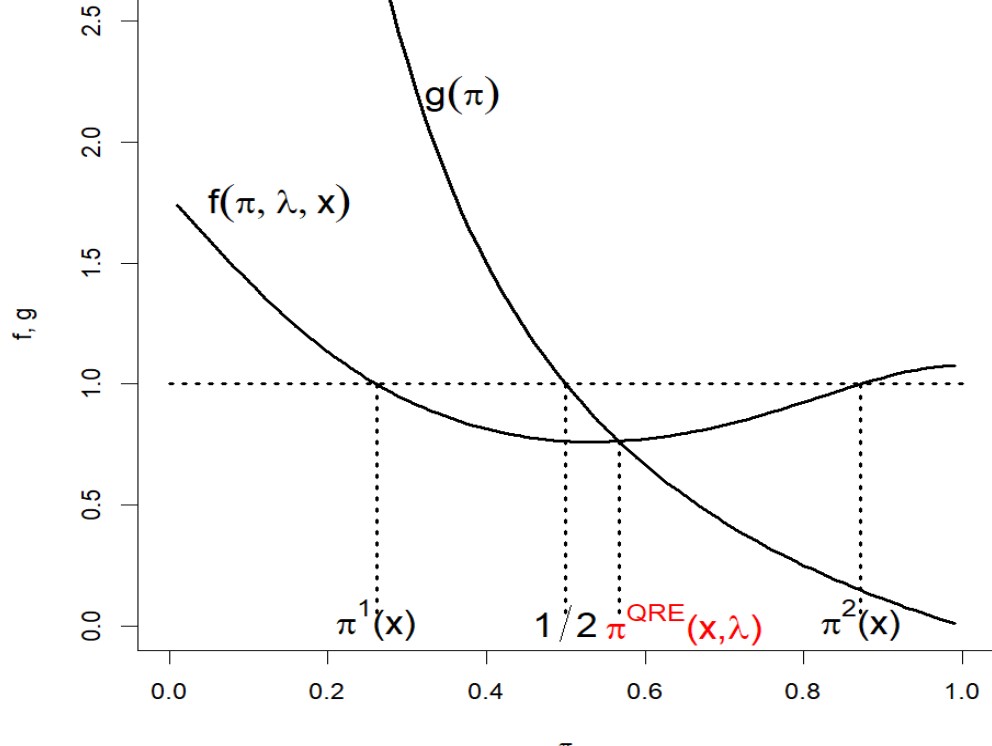

**Figure 2.** QRE in the case $\pi^1(x) < \frac{1}{2} < \pi^2(x)$. Parameters of the figure: $\lambda = 0.5, x = 1, n = 5$, $k = 3, c = 0.15, G = 1$.

In the CPG game, if there is no mixed strategy equilibrium for $x = 0$, then $\pi = 0$ is the only equilibrium and CP is unnecessary. If there are two equilibria for $x = 0$, then, according to Propositions 1 and 2, we are in a case where $\pi = 0$ is selected or in a case where the larger equilibrium $\pi^2$ is selected. As shown above, $\pi^2$ always increases with increasing $x$. Subsequently, CP is unnecessary or disadvantages for the principal.

**Conclusion 1.** *Relying on equilibrium selection by limits of QRE, in the CPG game, CP never increases the probability of complying.*

In the CPB game, the principal can sometimes increase the probability of compliance by CP; but, there are no simple conditions for the possibility and the extent of improvements. Let us discuss two somewhat extreme cases, namely rather small $c/G$ ratios and rather large $c/G$ with the restriction $M \le -G$. The latter excludes punishment levels surpassing the negative externality from frequent resistance.

For small enough $c/G$, we have a situation, as in Figure 1, with the rotation point below the curve $q(\pi)$ and two mixed strategy equilibria for $x = 0$. $\pi^1$ is close to 0 and $\pi^2$ is close to 1, i.e., $\pi = 1/2$ is between $\pi^1$ and $\pi^2$. $f(\pi, \lambda, x)$ is larger than 1 between $\pi^1$ and $\pi^2$ and Proposition 2 implies that $\pi^1$ is selected. For increasing $x$, $\pi^1$ decreases. For $x > x_0$ (necessarily $x_0 > -c$), there is no longer a mixed strategy equilibrium $\pi^1$ (the largest of the possibly two intersections of $q(\pi)$ and $r(\pi, x)$) but only the pure strategy equilibrium $\pi = 0$ and the mixed strategy equilibrium $\pi^2$. Proposition 2 implies that then $\pi = 0$ is selected.

For large enough $c/G$, there is no symmetric mixed strategy equilibrium for $x < -c$ (in Figure 1, $r(\pi, x)$ lies above $\sigma(\pi)$ and $\pi = 1$ is the only equilibrium). For $x > -c$, there is a mixed strategy equilibrium $\pi^*$, but, for $x < -G$ still $\pi = 1$ is selected. Only for $x \gg -G$, we may find $\pi^* > 1/2$ and the selection of the pure strategy equilibrium $\pi = 0$.

For intermediate $c/G$, the equilibria are more diverse and equilibrium selection may be more complex. For all values, Propositions 1 and 2 determine the selected equilibrium. Therefore, we can, for other given parameters, determine the optimal $x$ numerically.

**Conclusion 2.** *Relying on equilibrium selection by limits of QRE, in the CPB game, we find: For small enough $c/G$, $x = M$ is the principal's optimal choice connected with a probability $\pi^* > 0$ of resistance. For large enough $c/G$, $\pi^* = 1$ is the selected equilibrium except in a case with $M > -G$ and $x$ so large that $\pi^* > 1/2$. This need not be fulfilled even for $x \to \infty$, but if it is fulfilled, then $\pi^* = 0$ is selected.*

### 3.2. The Linear Tracing Procedure

Harsanyi and Selten [28] mainly select equilibria by the application of the *Linear Tracing Procedure*. The procedure starts from the *centroid* of a game $\Gamma$, which consists of the strategy profile, where every pure strategy is played with the same probability. In games with binary pure strategies, both strategies are played with probability $\frac{1}{2}$. In the tracing procedure, for every $0 \le t \le 1$, equilibria are determined in a game $\Gamma(t)$ where player $i$ assumes that, with the probability $t$, the game $\Gamma$ is played, and with $1 - t$ the other players decide according to the probabilities of the centroid. For symmetric games, only symmetric equilibria are considered. If there is a unique continuous path of equilibria from $t = 0$ to $t = 1$, then the equilibrium at $t = 1$ is selected. If there are different paths with different equilibria at $t = 1$, then Harsanyi and Selten [28] suggest additional criteria for the selection of a unique equilibrium.

**Proposition 3.** *Let us regard a symmetric game $\Gamma$ with a smallest and a largest strictly mixed strategy equilibrium, $\pi^1 \le \pi^2$. A player's marginal utility if all other players play $\pi = 1/2$ is $Q(0) = U_i'\left(\frac{1}{2}\right)$. The Linear Tracing Procedure selects the following equilibria.*

1. *If $Q(0) < 0$ and $\pi = 0$ is an equilibrium, i.e., $U'(0) < 0$, then $\pi = 0$ is selected.*
2. *If $Q(0) > 0$ and $\pi = 1$ is an equilibrium, i.e., $U'(1) > 0$, then $\pi = 1$ is selected.*

3. If $Q(0)\langle 0,\ U'(0)\rangle 0$ and $U_i'$ is non-increasing in the interval $(0, \pi^1)$, then $\pi = \pi^1$ is selected.

4. If $Q(0) > 0,\ U'(1) < 0$ and $U_i'$ is non-increasing in the interval $(\pi^2, 1)$, then $\pi^2$ is selected.

**Proof.** $i$'s marginal utility in the game $\Gamma(t)$ is

$$Q_i(t) = (1 - t) * Q(0) + t * U_i'(p), \qquad (14)$$

For 1 and with $p_i = 0$ for all players, $Q_i(t) < 0$ for all $t$, i.e., $\pi = 0$ is an equilibrium for all $t$. For 2, $\pi = 1$ is an equilibrium for all $t$.

For 3, $p_i = 0$ is not an equilibrium of the original game. For small $t$, $p_i = 0$ is a best reply for all players, but there is $t = t'$ with

$$Q(t') = (1 - t') * Q(0) - t' * U_i'(0) = 0, \qquad (15)$$

With increasing $t$, increasing symmetric strategies $p_i = \pi$ with decreasing $U_i'(\pi)$ keep $Q(t') = 0$. For $t = 1$, we get $Q(1) = U_i'(\pi^1) = 0$. The reasoning for 4 is the same, we only move from $\pi = 1$ to $\pi = \pi^2$. $\square$

In the CPG game, we are in the same situation as under the QRE selection: either $\pi = 0$ or $\pi^2$ is selected and $\pi^2$ decreases with $x$. Therefore, again, either CP is unnecessary or disadvantages the principal.

**Conclusion 3.** *Relying on the Linear Tracing Procedure, in the CPB game, CP never increases the probability of complying.*

In the CPB game, let us repeat the distinction from Conclusion 2. For small enough $c/G$ and $x < -c < -G$, we have $Q(0) < 0$ and $U'(0) > 0$. The situation is as in Figure 1 with the rotation point below the curve $q(\pi)$ and two mixed strategy equilibria for $x = 0$. Because of Proposition 3, $\pi^1$ is selected, which decreases with increasing $x$. Therefore, for $M < -c$, the principal will choose $x = M$, connected with a small but positive resistance probability. In the CPB game with $M > -c$, the principal can choose $-G > x > -c$. Subsequently, the complete compliance equilibrium $\pi = 0$ exists and it is selected (Proposition 3).

$U'(1) > 0$ is always fulfilled. For large enough $c/G$ and small $x$, no mixed strategy equilibrium exists (which implies $Q(0) > 0$) and the unique equilibrium $\pi = 1$ applies. Possibly, for $x$ sufficiently large (at any case $x > -c$), we get $Q(0) < 0$ and the selection of the then existing equilibrium $\pi = 0$.

As in the discussion of the QRE selection, for medium $c/G$, many different subcases have to be distinguished.

**Conclusion 4.** *Relying on equilibrium selection by the linear tracing procedure, in the CPB game, we find: For small enough $c/G$ and $M < -c$, the principal's optimal choice is $x = M$, connected with a probability $\pi^* > 0$ of resistance. For $M > -c$, the principal chooses $x > -c$ connected with $\pi^* = 0$. For large enough $c/G$, $\pi^* = 1$ is the selected equilibrium except in a case with sufficiently large $M > -c$ and provided $x$ exists so that $Q(0) < 0$. For such an $x$, $\pi^* = 0$ is selected.*

### 3.3. Comparison and Summery

For CPGs, the conclusions from the two equilibrium selection methods coincide: CP never decreases the probability of resistance.

For CPBs, the conclusions from the two methods qualitatively coincide. Both of the methods have many subcases, which are, in principle, covered by Propositions 1, 2, and 3. For every combination of parameters, we can find the equilibria and decide which equilibrium is selected. However, simple general rules are lacking. Therefore, the discussion is concentrated on the extreme cases of "sufficiently" small or large ratios $c/G$. For small

ratios, in both cases, $x = M$ would be connected with the smallest possible probability of resistance. However, the linear tracing procedure predicts smaller probabilities of resistance ($\pi^1$ or even $\pi^* = 0$ instead of $\pi^2$). For large ratios $c/G$, in most circumstances, complete resistance $\pi^* = 1$ cannot be avoided. Both of the procedures point out a possible exception with complete compliance, $\pi^* = 0$.

## 4. Discussion and Conclusions

Collective punishment takes many forms and the games CPG and CPB cover only part of these cases. International sanctions against a country (Iran, Russia, and others) do not have the required structure because there is only one agent acting. Additionally, many examples of hostage taking do not have the required structure. These are cases as described in the introduction, where the "culprits" are not available for punishment, but their people (international sanctions) or the hostages that they gave are. In addition to such complete mismatches, details of the analyzed models may be changed: the production of a "normal" instead of a threshold public good, other triggers for CP, positive or negative costs of CP for principals, asymmetric agents, extended dynamics, etc. Beauty is in the eye of the beholder, but I think that my choices of model specifications are, in many cases, appropriate under the aspects of plausibility and simplicity.

The Linear Public Good model and the (linear) Common Pool model are strategically equivalent, the CPG and the CPB are not. The reason is that, for strategic equivalence, also the sign of punishment has to be switched. In the CPB model, the principal punishes actions, which support the production of the public good. By exchanging the labels for the actions "resisting" and "complying", we get a model where "enough complying" produces a public good, which can be interpreted as the non-production of a public bad. For strategic equivalence, punishment has to be attached to the newly labeled compliance. This is equivalent to rewarding the newly labeled resistance. Therefore, we get strategically equivalent games only if we switch the signs not only of $c$ and $G$, but also of $x$. This is the reason why CPG and CPB games have different equilibria also after a re-labeling of strategies.

We have not entered an explicit normative discussion of CP. The introduction of a maximal punishment $M$ may be regarded as a consequence of such a discussion. Cutting off a hand of all potential thieves is certainly not acceptable, not even under the Sharia law. In the CPB game, it seems that large $M$ is more plausible when the gains of the agents $-c$ are small when compared with the damages $-G$ from a public bad than in the opposite case. For example, soft-speaking with neighbors in the classroom (without the identification of culprits) may be tolerated because $c/G$ is large (a small loss of welfare) and the, if existent, necessarily high $x$ seems to be too harsh. Fishing with dynamite, however, has a small $c/G$ ratio (high loss of welfare) and CP with $M > -c$ and, as a consequence, complete compliance is justified. If we follow this argument, then, in the CPB game, CP can and should be applied for small $c/G$, because it is effective and efficient. For large $c/G$, CP is mostly ineffective and promises, if at all, only small efficiency gains. Therefore, it should not be applied.

Which applications of the theoretical analysis are possible? For this purpose, we should match a suitable situation with CPG or CPB and estimate, in the latter case, whether the $c/G$ relation is high or low.

For the *CPG* game, the application of the model is simple if we have clear confrontation between an authority (for example, an occupier) and a group with similar goals (expel the occupier). Resistance, CP, and success are easily identifiable. Our advice for an occupier is: never threaten CP. We know from the history of armed conflicts that not every occupier followed this rule. A question to historians is whether there has been more or less resistance under the threat of CP than without. Protests against authoritative regimes seem to be similarly good examples, and we may conclude from anecdotal evidence that experienced CP and the threat of further CP never extinguish protests. The application of the model to such conflicts is plausible, but sometimes limited. The description of long-term conflicts

with many changing states by a static model may be regarded as an over-simplification. In addition, a closer look at public protests shows that we should avoid the mistake to believe that the protesters represent the whole population. Demonstrations for minority rights or against strict anti-abortion laws, etc., only express the opinion of a more or less large part of the population. Western observers generally agree with the requirements of Belorussian, Russian, and Hong Kong demonstrators, but a considerable part of the population supports the regimes. For many Russians, Putin's "Make Russia great again" policy has more weight than human rights.

In order to apply the CPG model to protests, we may concentrate on the group of demonstrators as the set of agents. All have the same goal, but only some are ready for violence, because they think, rightfully or not, that only violence produces attention and pressure. In this situation, the authority may try to separate the violent individuals (individual punishment), or they may apply CP with water guns and tear gas. We have a lot of anecdotal evidence about the application and non-application of CP in these situations, but, again, a detailed investigation would be necessary.

The *CPB* game has a less clear result. Successful policy depends a lot on the $c/G$ ratio. Football fans should not use pyrotechnic articles. Individual punishment is practically impossible, but neither is CP frequently applied. because, in this case, fans cause low damage $G$ when compared with their individual gains $-c$. Even vituperations of the opponent are tolerated up to a certain level, but violence and racism with large (negative) $G$, even for the average fan, are often followed by denied or restricted access of the fan community to the stadium.

CP plays an important role in the fight against organized crime. Often, it is clear that members of a certain group (gang, clan, family) have committed criminal acts, but individual responsibility is difficult to prove. In this situation, the application of laws like the Racketeer Influenced and Corrupt Organizations (RICO) statute in the US or the law against "Bildung krimineller Vereinigungen" in Germany allows for punishment based on the membership in or the support of criminal organizations. With the relatively low $c/G$ ratios caused by violent crimes, theft, and fraud, the model predicts the possible effectiveness of CP. Note that $G$ measures the general feeling of insecurity in society plus individual precautions (no-go areas, anti-theft measures, purchase of weapons, etc.). However, the success of RICO is questionable. While Giuliani [32] reports highly effective measures against criminal families in New York, Gabor [33] (Section 4.1), in a report for the Canadian government, claims the rare application of the law and its " ... "dramatic failure" in attacking OC's infiltration of legitimate businesses, ... " He regrets (Section 5) that "(m)uch of the evidence pertaining to the efficacy of OC control strategies is descriptive and anecdotal. Studies adopting sophisticated research designs are virtually non-existent".

Athletes rarely dope for their teams, but usually for their own advantage though, often, supported by their teams. Widespread detected doping is punished with the exclusion of teams, such as the exclusion of the Festina team from the 1998 Tour de France (followed by the "voluntary" surrender of six other teams) or Russia's ban from the Olympics and World Championships until 2022. The latter is one of the most severe cases of CP in a "civil" environment, i.e., excluding violence in wartime. All Russian athletes are prevented from being part of these summits of sport and winning their trophies. We may estimate that the negative externality of shaming a complete generation of athletes and the height of punishment are much larger than the individual advantage $-c$ of doping. According to the analysis of the CPB game, the now credible threat to exclude nations from the Olympics should induce broad compliance. However, I admit that such expectations might be exaggerated.

**Funding:** This research received no external funding.

**Institutional Review Board Statement:** Not applicable.

**Informed Consent Statement:** Not applicable.

**Data Availability Statement:** Not applicable.

**Conflicts of Interest:** The author declares no conflict of interest.

## Appendix A

**Proof of Lemma 3.** $\tau = \pi^2(0)$ is characterized by

$$q(\tau) = \binom{n-1}{k-1} * \tau^{k-1}(1-\tau)^{n-k} = \frac{c}{G}, \tag{A1}$$

$$\frac{\partial q}{\partial \pi}(\pi = \tau) \leq 0 \iff \tau \geq \frac{k-1}{n-1}, \tag{A2}$$

$\eta = 1 - \left(\frac{c}{G}\right)^{n-1}$ implies

$$\frac{c}{G} = (1-\eta)^{n-1}, \tag{A3}$$

From (A1) and (A3) follows

$$\binom{n-1}{k-1}\left(\frac{\tau}{1-\tau}\right)^{k-1} = \left(\frac{1-\eta}{1-\tau}\right)^{n-1}, \tag{A4}$$

From (A2) follows

$$\binom{n-1}{k-1}\left(\frac{\tau}{1-\tau}\right)^{k-1} \geq \binom{n-1}{k-1}\left(\frac{k-1}{n-k}\right)^{k-1} = \frac{(n-k+1)*\dots*(n-1)}{1*\dots*(k-1)}\left(\frac{k-1}{n-k}\right)^{k-1} > 1, \tag{A5}$$

and, therefore $\eta < \tau = \pi^2(0)$. □

**Proof of Proposition 1.** For the extreme probabilities we find

$$U_i'(0) = -x - c \text{ and } U_i'(1) = -c, \tag{A6}$$

$$\frac{\partial U_i'(\pi)}{\partial \pi}(\pi = 0) = (n-1)x, \quad \frac{\partial U_i'(\pi)}{\partial \pi}(\pi = 1) = 0, \tag{A7}$$

For sufficiently small $\varepsilon$,

$$U_i'(1-\varepsilon) > (<) -c \text{ if } x + G > (<)0, \tag{A8}$$

1. follows from (A6). For 2 and 3 we have to investigate the function $U_i'(\pi)$ more closely. We compute

$$\frac{\partial U_i'(\pi)}{\partial \pi} = (1-\pi)^{n-2}(x+G) * \left[h(\rho) + (n-1)\frac{x}{x+G}\right], \tag{A9}$$

with

$$h(\rho) := \binom{n-1}{k-1} * \rho^{k-2} * [(k-1) - (n-k)\rho], \tag{A10}$$

$$\rho = \frac{\pi}{1-\pi}, \tag{A11}$$

As we have assumed $k > 2$, $h(\rho)$ is an unimodal function with $h(0) = 0$, $h(\rho)$ positive for $0 < \rho < (k-1)/(n-k)$ ($0 < \pi < (k-1)/(n-1)$) and negative for $\rho > (k-1)/(n-k)$. For $\pi \to 1$ ($\rho \to \infty$), $h(\rho) \to -\infty$. Therefore, $(\partial U_i'(\pi)/\partial \pi = 0$ has exactly one solution for $x + G > 0$ and no or two solutions for $x + G < 0$. Therefore, $U_i'$ has one local extremum for $x + G > 0$ and zero or two local extrema for $x + G < 0$. Together with (A6)–(A8), this characterizes the equilibria.

2.  Because of $0 < c < G$, we have no or two mixed strategy equilibria. In the case without a mixed strategy equilibrium, $\pi = 0$ is the only equilibrium. In the case with two equilibria, $U_i'(\pi)$ increases from $\pi = 0$ to the lower equilibrium $\pi^1$ and it decreases from the higher equilibrium $\pi^2$ to $\pi = 1$.

3.  For $G + x < 0$ and $c + x < 0$, we have zero or two local maxima of $U_i'$. Considering the values for $\pi = 0$ and $\pi = 1$ and the derivative at $\pi = 0$ we find either no or two equilibria, together with the described signs of $U_i'$. For $G + x < 0$ and $c + x > 0$, the conditions at $\pi = 0$ and $\pi = 1$ allow one or three equilibria together with the described signs of $U_i'$. Exactly one equilibrium exists for $G + x > 0$ and $c + x > 0$. Note that this case is different from 2. because of $c < 0$. □

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
