# Peer review of "Deterrence by Collective Punishment May Work against Criminals but Never against Freedom Fighters"

_games, doi:10.3390/g12020041_

Round 1
Reviewer 1 Report
Strongly motivated by a metaphor for real world, this work proposes a new version of collective punishment game, which mainly consists of two versions: CP against a Public Good and CP against a Public Bad. The authors analyzed the equilibria in CPG and CPB game, which confirm the best response when people face different scenarios. Thus, I have basically a positive feeling for publication. Yet, I would like to suggest several points as below to improve the MS, which makes their work more impressive to the audience.
Collective punishment is universal in our society, and authors provided sufficient example to strengthen the argument, but I still recommend give more discussions about CP and basic punishment or reward, especially from the perspective of experiments. Add the further discussion by consult with some previous work as below; (i) Wang et al.; Exploiting a cognitive bias promotes cooperation in social dilemma experiments Nature communications 9 (1), 1-7, 2018, (ii) Li et al.; Punishment diminishes the benefits of network reciprocity in social dilemma experiments, PNAS 115 (1), 30-35, 2018. Here also have many literatures studying collective human behavior and threshold public good game which is similar to the collective punishment, authors can give some discussions in conclusion part, such as, (i)Wang et al., Communicating sentiment and outlook reverses inaction against collective risks, PNAS, 117 (30), 17650-17655, 2020. (ii)Andrews et al., High-risk high-reward investments to mitigate climate change, 8(10), 890-894, 2018
The figures are both interesting, but the authors should use high resolution version, and the figure caption should be more self-contained.
The manuscript could be more readable if the authors can pay close attention to the wordings as well as the formats, especially the Latex mathematical style. For instance, in line 185-186. There are some mistakes in the formula, for example, it should be ????(?(?−{?})=0) in Eq. (8), Please check the manuscript again.
I would suggest the authors to consider the above suggestions and make a revision of their manuscript.
Author Response
see the acttachment.

Reviewer 2 Report
This papers develop a model in which a principal can use collective punishment (CP), in the sense of establishing a punishment that affects all individuals in a group of agents, to deter them from performing undesirable actions. In Stage 1, the principal chooses the level of punishment to be triggered if at least one the agents do not comply with (unspecified) regime of behavior. In Stage 2, there are two different cases. In the CP against a Public Good (CPG) case, the agents can either resist, paying a cost, to produce a public good if the stock of resistance is high enough, or comply and get zero. In the CP against a Public Bad (CPB) case, agents can either resist, receiving a benefit, or comply and get zero, but if the stock of resistance is high enough a Public Bad is produced. The author claim that CP cannot induce compliance in the first case, but it can in the second one. Assessment Starting with the introduction, I think it could be better structured. For instance, I lack an early definition of CP, since the term could be interpret as a collective performing the punishment action, such as some social stigma directed to criminals, for example, or as a punishment that is applied to a collective as a whole, such as punishing a whole class for the actions of a rogue student. The problem analyzed in this work is the latter one, which is apparent by reading the examples, but stating the definition would be very useful. At the same time, I think more effort should be devoted to follow a classical structure of Motivation⇒Research question⇒Place in the literature⇒Relevance, and maybe some description of the model/methods. Other structures are possible, but in many cases the simpler, easy to follow one can help to better establish the position of the contribution in the literature, that here is very difficult to get. When we turn to the analysis of the model, several problems arise. In Stage 1, the principal determines the trigger and the level of the CP, x∈[0, M], where the trigger for CP is just one resistant agent and the objective of the principal is just to maximize compliance. However, no utility for the principal is specified, so that, if the objective is just to maximize compliance, and there is no cost on providing punishment, what impedes the principal to just establish the most severe level of punishment and that’s all? The incentives of the principal should be better specified, perhaps by thinking of a planner trying to maximize surplus of the society, or a planner that do not care about the welfare of the agents, but suffers an additional cost if the Public Good/Bad is produced. Then, we turn to the second stage, in which agents decide whether to resist, at a cost (benefit) c for the Public Good (Bad) case, and then, if a threshold k of agents is reached, they receive the personal net benefit (cost) G=G′−x of the Public Good (Bad) produced. The case of k= 1 reduces trivially to a simple volunteer’s dilemma, with the well known result that, the more agents there is in a group, the less likely is that any individual steps in. As stated by the author, this is independent of the level of CP. The problem is that it is not that apparent the effect of CP in the other cases. If k > 1, the problem of the Public Good production just becomes a threshold Prisoner’s Dilemma with multiple agents, so that the cost of cooperation is not c but c+x, although the problem remains the same. Provided that the cost after CP is high enough, CP would actually decrease resistance, although of course there would be parameter values in which the optimal mixed strategy assigns positive probability to resist. By the same argument, in the case of the production of the Public Bad, the problem reduces just to the one in which the personal benefit to resist becomes c−x, and hence there is sufficiently little punishment so that it is still individually profitable to resist. From then on, the claim of the paper of "CP doesn’t work in the CPG case, but it does in the CPB" is difficult to sustain, even when true under certain parameter values. At the end of the day, the problem is that the structure imposed by the author just scales the possible costs and benefits of resisting the principal, while not specifying the incentive structure of the principal.Author Response
see the acttachment.
